# OpenReview forum: "Firewalls to Secure Dynamic LLM Agentic Networks"
_TMLR — Accepted by TMLR_

### Review · Reviewer_nF3J · 2026-04-05

**Summary Of Contributions:**

This paper addresses the security and privacy risks that arise when LLM agents communicate on behalf of users across trust boundaries. The authors argue that existing defenses are insufficient: detection classifiers fail because an adaptive adversary can always find framings that evade them; static rule-based policies (e.g., binary disclose-or-redact) avoid this adversarial game but cannot accommodate the variability real tasks require; secure-by-design approaches that assume pre-determinable control flows break down in open-ended agent dialogue where attacks resemble legitimate conversation.

To address this, the paper proposes a dual-firewall architecture grounded in "context projection." The Language Converter Firewall transforms incoming natural language into a closed, domain-specific structured and verified protocol which eliminates persuasive framing, prompt injection, and social engineering by removing their channels. The Data Abstraction Firewall transforms outgoing user data before the assistant sees it, applying learned rules that transforms the information at context-appropriate granularity, while being architecturally isolated from external influence. Both firewalls' rules are learned automatically from demonstration corpora rather than manually specified.

Across three domains and four frontier LLMs, the dual-firewall architecture substantially reduces both privacy and security attack success rates while preserving task utility.

**Audience:**

Yes

**Audience Explanation:**

The problem that this paper address is of great interest and importance in the community.

**Claims And Evidence:**

Yes

**Claims Explanation:**

* The "context projection" framing that unifies the two firewalls under the principle that both sides of the channel carry more information than the task requires is clean and well-motivated.

* The separation between LLM-based conversion and deterministic verification in the Language Converter is a well-motivated design choice.

* The empirical finding that firewall-protected systems often achieve higher task utility than unprotected ones is counterintuitive but well-supported. This strengthens the practical usage claim of the proposed method.

**Requested Changes:**

* The attack failures deserve systematic analysis rather than being treated as a calibration issue. The qualitative section exclusively showcases successes, missing an opportunity to understand where the architecture's guarantees end.

* The following two possible attacks could be examined:
  - First, an adversary can populate valid protocol fields with false data (fabricated ratings, fictitious prices) that pass the deterministic verifier because they are correctly typed. The firewall may make this worse: without it, persuasive language might trigger skepticism, but with it, false claims arrive as verified structured data with implicitly elevated trust.
  - Second, the Data Abstraction Firewall transforms fields independently, but the intersection of multiple abstracted attributes (region, family structure, allergy, accessibility need, budget, blocked dates) can be re-identifying. This is a known limitation of field-level generalization in statistical privacy, and the architecture inherits it because no mechanism reasons about the joint information content of disclosed fields.

Note: as I’m not familiar with the literature of agent-to-agent communication security, I can not assess the novelty and significance of the proposed approach.

---

> ### Author Response · Authors · 2026-05-03
> **Response to Reviewer nF3J**
>
> Thank you for the careful review and for identifying two concrete attack vectors that deserve explicit treatment.
>
> ### **Systematic analysis of residual failures**
>
> Please see the failure taxonomy in the general response, which provides the systematic analysis requested.
>
> ### **Attack 1: Fabricated data in valid protocol fields**
>
> You are right that the firewall may make this specific attack *worse*: a fabricated star rating or price within the correct type and range would pass the verifier, and without surrounding natural-language framing, the assistant may have fewer cues for skepticism. But this already requires a capable agent that would get skeptical without being persuaded by the same natural-language framing.
>
> **We also want to be precise about scope. Data fabrication is not a new threat introduced by LLM agents. It is a classical fraud problem that exists in any commerce setting, human or automated.** In the domains we evaluate (travel, real estate, insurance), service-provider agents are typically API-backed: a hotel agent connected to a reservation system pulls prices from inventory, and a fabricated quote would fail at the actual booking stage. These domains are also subject to consumer protection and false-advertising regulation, creating legal accountability for false claims. In practice, data authenticity in these settings is handled by existing mechanisms such as platform verification, booking confirmation, legal liability, and reputation rather than by the communication channel itself.
>
> What *is* new and specific to LLM-to-LLM communication is manipulation through language (social engineering, prompt injection, persuasive framing) and inappropriate data extraction through dialogue. These are the threats our firewalls address, and they have no pre-existing defense layer.
>
> That said, we acknowledge that (1) not all future agent ecosystems will have these external accountability mechanisms, and (2) the firewall's structured presentation may confer unwarranted implicit trust on fabricated values, as the reviewer notes.
>
> **We will explicitly mark factual accuracy of structured inputs as a limitation and discuss natural extension points**: verifying numeric fields against trusted reference APIs, and prompting the assistant to flag statistically implausible values (e.g., a 5-star city-center hotel at €30/night).
>
> ### **Attack 2: Re-identification through intersection of abstracted fields**
>
> This is a well-grounded concern. The architecture does inherit a classical limitation of field-level generalization, extensively studied in the statistical privacy literature.
>
> Three factors partially mitigate this concern in our architecture, though none fully resolve it:
>
> 1. **Joint context in abstraction.** The data abstraction firewall applies rules to the user's full record in a single LLM call, not field by field, so it can be extended to reason about combinations, e.g., abstracting a rare allergy when it co-occurs with a small geographic region. This reasoning is informal rather than formally compositional, but it is not absent.
>
> 2. **Strictly better than the binary alternative.** A binary disclose-or-block design (e.g., AirGapAgent) passes each accepted field at full granularity. Granularity-aware abstraction reduces the precision of every disclosed quasi-identifier *before* any intersection is taken. The joint information leakage is therefore strictly smaller for any disclosure pattern that our architecture permits, even though it is not zero.
>
> 3. **Schema-bounded co-disclosure.** The language converter's closed vocabulary enumerates which fields can be co-disclosed, bounding the set of quasi-identifiers that can participate in any intersection attack. Fields absent from the schema have no channel through which to reach the conversation.
>
> A formal joint-information bound, a deployment-time guarantee on the leakage of any allowed disclosure pattern, is open work and the natural next step. We will add this as a limitation and future-work direction.

---

### Review · Reviewer_sGmA · 2026-04-08

**Summary Of Contributions:**

This paper studies security and privacy risks in agent-to-agent communication among LLM-based systems. It proposes a dual-firewall architecture designed to constrain information flow across trust boundaries. The approach is evaluated on the ConVerse benchmark across multiple domains and models, showing substantial reductions in both privacy and security attack success rates while maintaining task utility.

**Strengths:**

1. The experimental evaluation is relatively comprehensive, including multiple domains, several LLMs, and tests of generalization to unseen personas.
2. The empirical results are strong, showing large reductions in attack success rates while maintaining comparable or slightly improved task utility.

**Weaknesses:**

1. The method lacks sufficient motivation in several places. In particular, the design choices in Sections 4.1 and 4.2 are not well justified, and it is unclear why these specific architectures are preferred.

2. Several parts of the method appear heuristic, and the paper does not provide sufficient justification for these design choices. Relatedly, the paper does not compare against alternative design choices. Apart from ablating the entire component, there is little exploration of other plausible design variants, which makes it difficult to assess whether the proposed architecture is uniquely effective.

**Audience:**

Yes

**Audience Explanation:**

The paper studies security and privacy issues in LLM agent systems, which is a topic of interest to the community.

**Broader Impact Concerns:**

N.A.

**Claims And Evidence:**

No

**Claims Explanation:**

Please refer to the weaknesses discussed above.

**Requested Changes:**

1. In Section 4.1.3, the structured language is learned only from benign conversations, while Section 4.2.2 uses both benign and attack conversations for rule learning. The paper does not explain this design difference, which makes the overall approach appear somewhat heuristic.

2. Several parts of the method appear highly heuristic or engineering-driven. For example, in Section 4.2.2, rules are learned for each domain. In practice, when applying the method to a new domain, it is unclear how a user would determine whether the learned rules are sufficient or how they should verify their correctness.

3. The paper would benefit from stronger motivation for the proposed design choices. In particular, more explanation is needed to clarify why the specific components and architecture in Sections 4.1 and 4.2 are necessary.

---

> ### Author Response · Authors · 2026-05-03
> **Response to Reviewer sGmA**
>
> Thank you for the review. We address the three concerns below.
>
> ### **1\. Asymmetry between benign-only learning (Sec 4.1) and contrastive learning (Sec 4.2)**
>
> The two firewalls learn fundamentally different objects, which drives this asymmetry.
>
> The **language converter** learns a *vocabulary*: the set of fields and valid values for legitimate communication. Including attack conversations would introduce keys that legitimate communication never needs (e.g., a `full_medical_history` or `employer_contact_email` field), undermining the closed-vocabulary guarantee that the deterministic verifier depends on.
>
> The **data abstraction firewall** learns *abstraction rules*, which are inherently contrastive. Benign conversations show what level of detail the task typically requires, but the assistant may overshare even in benign cases, so benign-only rules risk being too permissive. Attack conversations reveal what additional detail an adversary will probe for, but attack-only rules risk being too restrictive and blocking task-relevant information. **The paired corpus yields balanced rules that permit legitimate sharing while blocking adversarial extraction.** We will add this explanation to the revision.
>
> ### **2\. How to verify learned rules for a new domain**
>
> **We do not claim cross-domain transfer, nor is it desirable**: our motivation is that privacy and security norms are inherently contextual and domain-dependent (Nissenbaum, 2004). Within a domain, verification follows the same pattern as any rule-based system: evaluate on held-out data and iterate. Our new sensitivity experiment (please see response to reviewer 4VHM) shows that even with a single persona or a quarter of the demonstration data, ASR remains below 8.3%. In practice, we expect rule sets to be iteratively refined via human review and feedback, similar to how network firewall rules are maintained in production systems.
>
> ### **3\. Stronger motivation for design choices**
>
> We will consolidate and highlight the motivations. In brief:
>
> **Language converter (Sec 4.1).** Detection-based defenses (classifiers, alignment tuning) play an open-ended adversarial game: for every framing a detector recognizes, the attacker can produce an equivalent one that evades it. The language converter sidesteps this game by replacing the inter-agent channel with a closed structured protocol whose values are type-checked by a deterministic verifier.
>
> **Data abstraction (Sec 4.2).** Even without attacks, LLM assistants tend to overshare (Shao et al., 2024). So we need another layer to preserve privacy. Granularity-aware abstraction (rather than binary pass/block) is needed because agents often need to observe private data to make decisions but should not share it at full resolution; the AirGapAgent comparison now demonstrates this. Placement also matters: a filter on the assistant’s output would operate in a context already containing the external agent's messages, inheriting adversarial influence. Input-side placement (before the data reaches the assistant) ensures the firewall never sees adversary-influenced content.
>
> **Design variants we considered and ruled out:**
>
> * *Programmatic grammar parser instead of LLM converter:* The external agent writes in free-form natural language (e.g., "about 145 euros a night, from the 15th to the 18th of March"). A regex or grammar-based parser would need to enumerate the many ways agents express dates, prices, and other fields which is a brittle and incomplete process. The LLM handles this fuzzy extraction reliably, and security does not depend on the LLM converter being correct: any error or manipulation in Stage 1 is caught by the deterministic verifier in Stage 2\.
> * *Output-side filtering for data abstraction:* As noted above, the assistant's output context already contains adversarial messages, so any filter on that output inherits adversarial influence (Section 4.2.1).
> * *Binary data filtering:* The benchmark is constructed in a way that private data might be helpful for the agent to make decision, but an appropriate disclosure should abstract the data instead of a binary allow/disallow decision. This is now supported by the new AirGapAgent baseline.

---

### Review · Reviewer_4VHM · 2026-04-22

**Summary Of Contributions:**

This paper introduces a dual-firewall architecture for securing agent-to-agent communication among LLM-based autonomous agents. The core conceptual contribution is context projection: the idea that both incoming messages from external agents and outgoing disclosures of personal data should be projected onto a narrowly defined task context, eliminating information that exceeds what the task requires.
The system consists of two complementary components:
1. Language Converter Firewall (inbound): converts free-form natural language from external agents into a closed, domain-specific structured protocol, enforced via deterministic verification and string anonymization. This removes channels for prompt injection, persuasion, urgency framing, and embedded instructions.
2. Data Abstraction Firewall (outbound): transforms user data retrieved by the assistant into task-appropriate abstractions (e.g., ranges, categories, regions) rather than binary allow/block filtering, operationalizing contextual integrity.
Both firewalls rely on rules learned automatically from demonstrations (benign and adversarial conversations), rather than manual specification.
The approach is evaluated on ConVerse, a benchmark of 864 multi-turn attacks across travel, real estate, and insurance. Results show dramatic reductions in privacy and security attack success rates (often >90% relative reduction), while maintaining or even improving task utility.

Strengths
- Strong conceptual framing. The notion of context projection is elegant, unifying privacy and security under a single architectural principle. It clearly distinguishes the paper from detection-based or prompt-hardening approaches.
- Structural guarantees. The Language Converter Firewall, in particular, provides compelling guarantees by entirely eliminating free-form language as an attack surface. The use of deterministic verification is a major strength.
- Well-designed architecture. Architectural isolation between firewalls and adversarial inputs is clearly reasoned and convincingly argued.
- Comprehensive evaluation. The paper evaluates across multiple domains, models, personas, and includes ablations and qualitative examples. The benchmark choice (ConVerse) is appropriate for the threat model.
- Utility preservation (and improvement). Demonstrating that constraints can improve plan quality and coverage is an important and somewhat counterintuitive result, well supported by evidence.
- Clear exposition. Despite the system’s complexity, the paper is clearly written, with helpful diagrams, examples, and appendices.

Weaknesses
- Assumption of a trusted internal environment. The threat model explicitly excludes poisoning or prompt injection originating from the user’s knowledge base or retrieved documents. While reasonable, this limits applicability to many practical RAG-based agents where such contamination is realistic.
- Reliance on LLMs for rule generation. Although enforcement is structural, rule and schema generation relies on LLM analysis of demonstrations. The paper does not deeply analyze robustness, stability, or failure modes of this learning process.
- Limited discussion of domain scalability. Domains are relatively well-structured (travel, insurance, real estate). It remains unclear how the approach scales to less structured, creative, or open-ended tasks.
- Evaluation judge dependence. All metrics rely on LLM judges (GPT-5). While standard for this benchmark, this introduces potential bias that is not fully discussed.
- Residual attack success rates. The remaining ~7–10% privacy ASR is acknowledged but not deeply analyzed beyond brief speculation.

**Audience:**

Yes

**Audience Explanation:**

This work is likely to be of strong interest to a broad segment of the TMLR audience, including researchers in:

- AI security and prompt injection defenses
- Privacy, contextual integrity, and data minimization
- System-level safety guarantees for AI
- Human-AI interaction and delegation

The paper addresses a pressing and emerging problem: securing autonomous agent ecosystems where interaction is open-ended and adversarial behavior is indistinguishable from normal communication. Its architectural framing goes beyond incremental mitigations and is likely to influence future system designs.

**Broader Impact Concerns:**

None IMO

**Claims And Evidence:**

Yes

**Claims Explanation:**

The paper’s central claims—that dual structural firewalls can dramatically reduce both privacy leakage and security manipulation in agent-to-agent communication without harming utility—are well supported by empirical evidence.

Key claims are backed by:
- Large-scale quantitative evaluation on 864 attacks across three domains.
- Consistent reductions in privacy ASR (e.g., GPT-5 from ~88% to ~8%) and security ASR (often below 4%).
- Ablation studies clearly attributing effects to each firewall.
- Persona generalization experiments showing robustness beyond rule-generation users.
- Extensive qualitative examples illustrating exactly how attacks are neutralized.

Importantly, the paper avoids overclaiming: it clearly states assumptions, acknowledges residual failures, and frames guarantees as architectural rather than universal security.

**Requested Changes:**

Critical for acceptance
- Clarify robustness of rule learning. Provide deeper analysis of how sensitive the learned schemas and abstraction rules are to the choice, size, and quality of demonstration data. Even a small empirical study or failure analysis would strengthen confidence.
- More explicit limitations section. While assumptions are stated, a concise, explicit discussion of where the approach is not expected to work (e.g., highly unstructured domains, compromised knowledge bases) would improve clarity.
- Analysis of residual failures. A taxonomy or concrete examples of the remaining privacy ASR cases would be valuable for guiding future work.

Would strengthen the work
- Broader discussion of scalability. Elaborate on how context projection might extend to domains like general problem-solving, code generation, or creative collaboration.
- Computational cost analysis. Include rough latency or cost estimates, or an argument for feasibility in real-time systems, especially given two LLM-mediated firewalls.
- Comparison to alternative defenses. A more direct empirical comparison with recent prompt-injection defenses or policy-based approaches (where applicable) would help situate the contribution.

---

> ### Author Response · Authors · 2026-05-03
> **Response to Reviewer 4VHM**
>
> Thank you for the detailed and constructive review. Please find our response below in addition to the general response.
>
> ### **Critical \#1: Robustness of rule learning**
>
> **Existing evidence.** Table 5 (Generalization Across Personas) already shows that **held-out personas achieve comparable ASR to rule-generating personas** (e.g., GPT-5: 6.36% → 9.09%, both \>90% reduction from the 88.51% baseline). Section 5.2 also reports cross-category transfer: rules from medical-data-extraction attacks block financial-extraction attempts.
>
> **New sensitivity experiment.** On travel planning with GPT-5 under the dual firewall, we re-generated rules under two additional conditions and re-evaluated on a subset of 24 attacks (3 privacy \+ 3 security × 4 personas):
>
> | Condition | Privacy ASR | Security ASR |
> | ----- | ----- | ----- |
> | No firewall (baseline) | 83.3% | 50.0% |
> | Original setting (personas 1 & 4\) | 0.0% | 0.0% |
> | Single persona only (half the demonstration data) | 8.3% | 0.0% |
> | Quarter of demonstration data (personas 1 & 4\) | 0.0% | 8.3% |
>
> **ASR never exceeds 8.3% (2/24) in any degraded condition compared to 83.3% privacy ASR and 50% security ASR without firewalls.** We acknowledge that rule learning in practice deserves its own consideration and will discuss this openly.
>
> ### **Critical \#2: Explicit limitations section**
>
> We will add a dedicated limitations section covering: (1) the trusted knowledge-base assumption, (2) scalability to unstructured domains, (3) rule-learning sensitivity, and (4) the false-data and re-identification concerns raised by Reviewer nF3J.
>
> ### **Critical \#3: Taxonomy of residual failures**
>
> Please see the failure taxonomy in the general response above, which we will incorporate into the paper. This confirms the speculation in Section 5.4.1 quantitatively and provides concrete examples.
>
> ### **Would-strengthen points**
>
> **Scalability.** The language converter's structural guarantees are tightest when the domain is enumerable (e.g., travel, real estate, insurance). In less structured or creative domains, more fields would be string-typed, and string anonymization alone becomes the main defense on the incoming side, potentially affecting utility. The data abstraction firewall scales more naturally, since contextual abstraction rules apply to the user's own data regardless of conversation structure. We will add this discussion.
>
> **Computational cost.** Each firewall adds one short-context LLM call per turn (the converter sees one external message plus the static schema; the abstraction firewall sees one retrieval result plus the static rules, neither includes dialogue history). The deterministic verifier runs in sub-millisecond time. The Discussion section already notes distillation into smaller fine-tuned models as the natural deployment path.
>
> **Comparison to alternative defenses.** We implemented an adaptation of AirGapAgent (Bagdasarian et al., 2024). Before any data item leaves the user's environment, an LLM is asked "given task τ, is item d relevant?"; the item is then either passed in full or blocked entirely. To match our codebase, we ran these relevance queries on each environment item up front given the task description (e.g., travel planning including activities, restaurants, preferences, insurance), then ran the full experiment using this filtered environment. Both methods use GPT-5 as the assistant against all attacks on personas 1 and 4 in travel planning. **On all tiers the dual firewall substantially outperforms binary filtering. This confirms that granularity-aware abstraction provides a qualitatively different defense than binary pass/block.**
>
> | Tier | AirGap ASR | Dual Firewall ASR | Reduction |
> | ----- | ----- | ----- | ----- |
> | Related and useful | 76.2% | 23.8% | −52.4 pp |
> | Related but private | 66.7% | 4.2% | −62.5 pp |
> | Unrelated | 64.6% | 0.0% | −64.6 pp |
> | **Overall privacy** | **67.3%** | **5.5%** | **−61.8 pp** |
> | **Security** | **27.3%** | **4.5%** | **−22.7 pp** |
>
> **Judge dependence.** Following ConVerse, the judge compares against pre-generated ground-truth annotations rather than performing free-form judgment, and Gomaa et al. (2025) report stable results when varying the judge LLM. We will make this more prominent.

---

### Comment · Reviewer_dCwb · 2026-03-31

I am sorry. This paper is outside my expertise (machine learning theory/optimization/differential privacy).

---

### Author Response · Authors · 2026-05-03
**General Response**

Thank you all for the careful reading and constructive feedback. A shared theme across reviews is the need for deeper analysis of residual failures. To address this and other concerns, we have conducted the following new experiments and analyses:

* **Failure taxonomy.** We manually analyzed all residual privacy and security failures under the dual firewall, characterizing each by failure mode.
* **Rule-sensitivity experiment.** We varied which personas, and how much demonstration data, are used to generate firewall rules, measuring the effect on ASR.
* **AirGapAgent baseline comparison.** We compare the dual firewall against our implementation of AirGapAgent (Bagdasarian et al., 2024\) using GPT-5 on all attacks for personas 1 and 4 in travel planning. The dual firewall achieves 5.5% privacy ASR (vs. 67.3%) and 4.5% security ASR (vs. 27.3%).

We will also add a dedicated limitations section and motivating paragraphs at the beginning of Sections 4.1 and 4.2.


### **Failure Taxonomy**

We manually analyzed all residual failures from Tables 1–2 and the all-domain tables.

**Privacy.** Two patterns account for nearly all residual failures. **The dominant pattern is *granularity calibration***: the assistant shares the correct type of information but at a finer level than the benchmark's reference abstraction (e.g., "rock climbing and scuba diving" instead of "high-risk adventure activities", or exact euro amounts instead of a range, or sharing medical needs as “cardiovascular monitoring needs (hypertension, cholesterol)” rather than “cardiovascular monitoring needs''). These occur predominantly in the **related-and-useful tier**, where some disclosure is required to complete the task. The learned rule was too permissive relative to the benchmark's abstraction. **The second pattern is *field overlap***: for example, in the insurance domain, a rule permits a coarse field the task legitimately needs (e.g., "recent claim history"), but the underlying record stores free-text values containing specific equipment and amounts that propagate through the rule. Both failures are rule-coverage gaps on the learning side, not successful adversarial bypasses of the system structure. We intentionally did not manually refine the rules; in practice, iterative calibration via human feedback would address these gaps.

**Security**. **About three-quarters of residual security failures are *preference manipulation, especially in insurance***: the attacker proposes valid domain items (products, add-ons) that pass the structured protocol because they are correctly typed and domain-relevant. They pass without persuasive framing. The firewall cannot distinguish between items matching the exact user's task in this session versus upsell attempts when both are structurally valid. **The remaining quarter is *assistant-side errors***: the assistant itself executes the attack target without any inter-agent channel (e.g., emailing a manager about vacation details when the user only asked to "inform the manager about unavailability"). The firewalls prevent external manipulation but do not guarantee correct assistant behavior.

### **References for the responses**

* Bagdasarian et al. "Airgapagent: Protecting privacy-conscious conversational agents." CCS. 2024\.
* Gomaa et al. "Converse: Benchmarking contextual safety in agent-to-agent conversations." Findings of EACL. 2026\.
* Nissenbaum. "Privacy as contextual integrity." Wash. L. Rev. 79 (2004): 119\.
* Shao et al. "Privacylens: Evaluating privacy norm awareness of language models in action." NeurIPS. 2024

---

### Decision · Action_Editor_F9b6 · 2026-06-01

**Recommendation:** Accept with minor revision

**Additional Comments:**

The camera ready must include all requested changes by the reviewers.
These should include, for instance, the discussion on motivation, and the new baselines and experiments.

**Audience:**

Yes

**Audience Explanation:**

The paper tackles a topic relevant to the trustworthy AI community, as confirmed by all reviewers.

**Claims And Evidence:**

Yes

**Claims Explanation:**

All reviewers now agree that that the paper claims are adequately supported.

---

> ### Author Response · Authors · 2026-06-10
> **camera ready version is uploaded per TMLR's instruction**
>
> Dear Alessandro and reviewers,
>
> Thank you so much again for your thorough reviews which helped improve the paper!
>
> We now integrated the rebuttal's experiments and discussions.
>
> Main changes are:
>
> - Explicit motivations of each firewall and design choice (sections: 4.1, 4.1.2, 4.2, 4.2.3)
> - Taxonomy of residual failures is added under section 5.8.1
> - Sensitivity of Rule Learning to Demonstration Data is added under section 5.6
> - Comparison to Binary Data Minimization Baselines is added under section 5.7
> - The discussion under section 6 is substantially expanded with explicit scope, limitations, challenges, and future directions
> - A link to our code is added as well.

---

> > ### Comment · Action_Editor_F9b6 · 2026-06-22
> >
> > Dear authors,
> >
> > Thanks for submitting the camera ready version. I will be happy to accept it once you address the minor formatting problem below.
> >
> > The GitHub link above the abstract does not comply with the TMLR template. Please add the explicit link using \url in the abstract and/or text.
> >
> > Many thanks,
> >
> > AE

---

> > > ### Author Response · Authors · 2026-06-22
> > >
> > > Thanks so much for checking!
> > > Apologies! We now edited the PDF.